# The Effectiveness of Virtual Reality on Anxiety and Performance in Female Soccer Players

**DOI:** 10.3390/sports9120167

**Published:** 2021-12-13

**Authors:** Kaitlyn Harrison, Emily Potts, Adam C. King, Robyn Braun-Trocchio

**Affiliations:** Department of Kinesiology, Texas Christian University, Fort Worth, TX 76129, USA; k.p.harrison@tcu.edu (K.H.); e.n.potts@tcu.edu (E.P.); a.king@tcu.edu (A.C.K.)

**Keywords:** penalty kick, relaxation, stress, VR

## Abstract

With the increased use of technology, relaxation interventions are finding their way into technology devices like virtual reality head-mounted displays (VR HMDs). However, there is a lack of evidence on the efficacy of VR relaxation interventions to reduce anxiety in athletes and how that is portrayed in their movement patterns. The purpose of the current study was to examine how a VR relaxation intervention affected perceived anxiety levels and penalty kick performance of female soccer players. Thirteen female soccer players took five penalty kicks in baseline, stress-induced, and VR relaxation conditions. Perceived levels of anxiety, self-confidence, mental effort, heart rate (HR), accelerometry of the lumbar spine and thigh, and performance in each condition was obtained. Results indicated that the VR intervention significantly reduced cognitive anxiety and somatic anxiety from baseline (*p* = 0.002; *p* = 0.001) and stress (*p* < 0.001; *p* < 0.001) with large effect sizes (Kendall’s *W* = 0.72; 0.83). VR significantly increased self-confidence from baseline (*p* = 0.002) and stress (*p* = 0.001) with a large effect size (Kendall’s *W* = 0.71). Additionally, all participants felt that VR helped them relax. Mental effort was significantly higher in the stress condition compared to that in baseline (*p* = 0.007) with moderate effect size (Kendall’s *W* = 0.39). Peak acceleration and performance were not significantly influenced by stress or VR. This study serves as an initial step to evaluate VR relaxation interventions on performance in female soccer players.

## 1. Introduction

Coaches, athletes, and practitioners are seeking strategies to optimize performance, and with the ever-evolving world of technology, virtual reality (VR) has seen an increase in use as a means to optimize performance. VR has been used to enhance perceptual-cognitive skills by training athletes to detect informational cues related to the game [1]. VR has also been utilized for injury rehabilitation in each phase of the rehabilitation process, tending to the needs of the participant during their recovery. Lastly, VR has been used for relaxation in sport to help athletes properly manage potential stressors by teaching athletes coping strategies with VR [1]. However, more information is needed with VR as a relaxation intervention for athletes under stress and competitive anxiety.

Stress and competitive anxiety are areas that have received considerable attention in athletics. Stress occurs when there is an imbalance between physical and mental demands placed on the athlete, and the response capability under those conditions fails to meet the demands, having important consequences [2]. More specifically, psychological stress is “a relationship between the person and the environment that is appraised by the person as taxing or exceeding [their] resources and endangering [their] well-being” [3]. A contributor to psychological stress is competitive anxiety [4]. Competitive anxiety is either a trait and/or state-like response to a stressful sport-related situation which results in a range of cognitive appraisals, behavioral responses, and/or physiological arousals. When it becomes too high, the athlete can become over-aroused, exceeding the mental capacity to process stimuli, resulting in an increase in stress and decline in performance [4]. Martens and colleagues [5] separate competitive anxiety into cognitive anxiety (e.g., negative thoughts and worry) and somatic anxiety (e.g., physiological signs of nervousness and tension). It has been demonstrated that negative coping control is associated with competitive anxiety [6]. Therefore, it is important for athletes to modulate psychological stress and anxiety by developing or discovering positive coping mechanisms for optimal performance and well-being [7]. Poor coping mechanisms indicate poor adaptability to stress. A negative reaction to stress and anxiety interferes with motor coordination, reduces flexibility, and increases cognitive and somatic anxiety, and shifting of attention leads to a narrowing of the visual field, causing athletes to miss vital cues [2,8].

Literature has demonstrated a negative relationship between anxiety and how it projects physically in high-anxiety situations [9,10,11]. A reduction in performance due to increased anxiety has been shown in soccer players taking penalty kicks, where participants performed significantly lower in the high-anxiety condition compared to the low-anxiety condition [11]. Sekiya and Tanaka [10] investigated the kinematic performances of novice table tennis players with high psychological pressure and found significant alterations in their swing and speed, contributing to a decrease in performance. Together, these studies show that increased stress and anxiety can negatively affect a person’s movement in multiple sports, indicating that stress can negatively impact not only psychological performance, but also physical performance. Because of this, it is important to consider both components to have a better understanding of stress and anxiety as a whole on athletic performance.

In order to assist with the increasing psychological stress and anxiety of athletes, various interventions have been implemented, including mindfulness, relaxation imagery, deep breathing, and muscle relaxation [7,12]. Now, with the increased use of technology, relaxation interventions are finding their way into technology devices and have become incredibly accessible with mobile apps (e.g., Calm and Headspace) and virtual reality head-mounted displays (VR HMDs). Due to the heightened immersive effects, VR HMDs can result in a better imagery experience and reduction of anxiety compared to traditional relaxation techniques [13]. VR has been used in other non-athletic populations (i.e., intensive care unit (ICU) patients, teachers, and nurses) for anxiety reduction, and a positive relaxation effect was found [14,15]. Exposure therapy with VR is a method that has been used to reduce the stressful reaction to specific stimuli with individuals to develop better coping mechanisms to reduce stress and allow them to apply their strategies in their professional setting [15]. For patients in an ICU, an audiovisual imagery experience was utilized to reduce sensation of stress and anxiety to put the patients at ease [14]. Furthermore, Liu and Matsamura [16] found that VR relaxation interventions provided acute relaxation in NCAA Division I student athletes. Researchers found that 74.4% of participants thought the VR intervention helped them relax or reduce anxiety, and 90% of the participants would use the intervention again [16]. This study did not analyze any performance factors and was solely focused on relaxation. Based on their post-VR survey, 67.5% of participants believed that the VR intervention could be beneficial in helping increase their performance when used before games, which opens a window for potential use of VR in sport that needs to be further explored in quantitative measures [16]. Thus, the current study aimed to provide evidence if VR relaxation could be beneficial prior to competition, or if it is a subjective perception among athletes.

Unfortunately, there is a lack of evidence on the efficacy of VR relaxation in anxiety reduction in athletes and how it translates to their movement patterns in performance settings. This study utilized accelerometer data as the physical component to examine if induced stress and/or the VR intervention affected participants’ technique/motor pattern and tracked any changes across the duration of the study. Due to the novelty of this study and it serving as an initial step in this direction of research, accelerometer data was utilized for its ease of use and simplicity in measuring biomechanical changes on a soccer field. Combining accelerometer data with heart rate (HR) and psychological measures will further bridge the sport psychology and biomechanical literature together to examine the effects of stress and anxiety as a whole. To extend the previous literature, the purpose of this study was to examine how VR relaxation techniques affect perceived anxiety levels in female soccer players and how the potential changes translated to their movement patterns during baseline, stress-induced, and VR relaxation penalty kick conditions. It was hypothesized that cognitive and somatic anxiety, perceived mental effort, and HR would be higher during the stress-induced block compared to those of the VR and baseline blocks, while self-confidence would be lower. It was also hypothesized that accelerometer data from the anterior thigh would be greater in the stress-induced block compared to those in the VR and baseline blocks and result in lower performance scores.

## 2. Materials and Methods

### 2.1. Research Design

Participants performed under baseline, stress, and VR conditions in a repeated measures design. Each participant served as their own control and completed all conditions. The current study followed similar procedures as those in Wilson et al. [11] and Wood et al.

### 2.2. Participants

Participants consisted of 13 healthy female soccer players, aged 19–22 (20.54 ± 1.127), and with at least two years of soccer experience. Out of the 13 participants, 11 participants were Caucasian, 1 was Asian, and 1 was African American/Black. Participants consisted of a goalkeeper (*n* = 1), defenders (*n* = 3), midfielders (*n* = 5), and forwards (*n* = 4). Overall soccer experience averaged 13 years (12.62 ± 3.62). Only one participant had prior experience using VR.

### 2.3. Instrumentation

#### 2.3.1. Demographic Questionnaire

A demographic questionnaire obtained information about the participants including items such as age, race, soccer history, and previous experience with VR. The VR items consisted of the use of VR as a relaxation technique and frequency.

#### 2.3.2. Mental Readiness Form-3 (MRF-3)

The MRF-3 measures cognitive anxiety, somatic anxiety, and self-confidence [17]. This instrument has three questions on a Likert scale from 1 to 11. Cognitive anxiety was assessed by rating thoughts about performance from 1 “being worried” to 11 “being not worried”. Somatic anxiety was assessed by rating physical manifestations from 1 “being tense” to 11 “being not tense”. Self-confidence was assessed from 1 “being confident” to 11 “being not confident”. Krane [17] developed the MRF-3 to address the concerns about the terms in the MRF-Likert being truly bipolar opposites and compared the results to those of the Competitive State Anxiety Inventory-2 (CSAI-2) with correlations of 0.76 for cognitive anxiety, 0.69 for somatic anxiety, and 0.68 for self-confidence. Krane [17] concluded that the MRF-3 is a suitable tool in the field of sport anxiety research due to its brevity and simplicity to complete the questionnaire and could be more advantageous to use than the CSAI-2 when facing time constraints. This form of the MRF has been used in a similar study by Wilson and colleagues [11] when assessing attentional control theory (ACT) in soccer players.

#### 2.3.3. Rating Scale for Mental Effort (RSME)

The RSME assessed the mental effort of participants invested in the penalty kick tasks. It is a one-dimensional scale on a vertical axis with a range from 0 to 150 [18]. There are descriptors on the scale with corresponding numbers to act as verbal references at 0 = not at all effortful, 75 = moderately effortful, and 150 = very effortful. Participants mark the scale according to their perceived effort on the task. This scale is reliable (0.88 in laboratory settings and 0.78 in real life settings) and a valid measure of mental effort [19,20].

#### 2.3.4. Kinematics

Kinematic data of the task were collected using the DELSYS Trigno Avanti (DELSYS Incorporated, Natick, MA, USA) sensors to record accelerometer data. Two sensors were used with one placed at the fifth lumbar/first sacral vertebrae, between the posterior superior iliac spines. The second sensor was placed on the anterior thigh at the midway point between the superior aspect of the patella at the knee and the anterior superior iliac spine at the hip of the kicking leg. EMGworks (DELSYS Incorporated, Natick, MA, USA) software collected data at 150 Hz. Accelerometer data was utilized to examine the physical translation of the stress inducer and VR intervention had on the participants to gather a wholistic view on the effects of stress and VR. The biomechanical literature has examined the effects of stress in skill execution but has yet to be combined with the sport psychology literature findings. Thus, this study aimed to combine these areas of the literature together to examine how the physical and psychological components interacted with each other with the induced stress and VR intervention.

#### 2.3.5. Virtual Reality Head-Mounted Display (VR HMD) and Application

The relaxation intervention played using the Oculus Quest (Facebook Technologies, LLC, Menlo Park, CA, USA) with a virtual relaxation session from the Liminal VR application (Liminal VR, Abbotsford, Victoria, Australia). The four-minute campfire screen from the calm category was used.

#### 2.3.6. Heart Rate (HR)

HR was obtained utilizing a Polar H10 sensor with a Pro Strap that sent data to the Polar Beat app via an iPad (Polar Electro Oy, Kempele, Finland). For each participant, HR was recorded prior to each penalty kick and then averaged to represent the overall HR for each condition. Gilgen-Ammann, Schweizer, and Wyss [21] tested the RR interval measurements of the Polar H10 sensor compared to those of the 3-lead ECG Holter monitor (Schiller Medizintechnik GmbH, Baar, Switzerland) that is referred to as the gold standard for HR data collection. RR intervals are the two consecutive R-waves in an electrocardiogram, and the signal quality of these RR intervals is what is important for measurement devices quantifying HR and HR variability [21]. Results showed that the Polar H10 was as accurate as the Holter monitor during low- and moderate-intensity activities and even had a higher RR interval signal quality than the Holter monitor during intense activities. Both systems had less than a 2% difference between each other in 97.1% of measured RR intervals and had a high correlation with each other (*r* = 0.997), thus, the Polar H10 monitor provides an accurate measurement of HR [21].

#### 2.3.7. Commitment Check

The Igroup Presence Questionnaire [22] checked the participants’ commitment. The questionnaire measures the sense of presence that individuals experience in VR. It consists of 14 questions divided among three subscales of spatial presence, involvement, and experienced realism. Questions were answered on a 7-point Likert scale from −3 to +3 (i.e., “fully disagree” at −3 and “fully agree” at +3, with 0 being neutral). Participants with a score lower than zero on the commitment check were excluded. The reliability of the spatial presence subscale is 0.80, the involvement reliability is 0.76, and the experienced realism subscale has a reliability of 0.68; overall, the IPQ has a reliability of 0.85 with all reliabilities using Cronbach’s alpha [22]. Igroup.org [22] conducted two studies to determine the IPQ’s reliability to determine VR presence along with a factorial analysis that can be referenced from their website.

Based on Lui and Matsumura [16], four additional questions were added to assess how relaxing the participants found the VR session. The questions were answered with a 5-point Likert scale ranging from “not at all” to “very much so.”

#### 2.3.8. Penalty Kick Scoring

Wood and Wilson’s [23] scoring zones were replicated in this study. The goal box was divided into twelve zones, with each half of the goal consisting of six zones of 61 cm, starting from an “origin” in the center (0 cm) and moving out to each post. Shots hit in the zones further from the central origin of the goal reflected shots that were further from the goalkeeper’s reach, which gave the participants higher scores. If the goalkeeper made a save, then the participant did not receive any points. If the participant completely missed, the participant lost five points. The zones were set with points in increments of five, thus, the four corners of the goal box had the highest points available. To determine where the shot was hit, a researcher marked where each shot was placed on a score sheet, and then scores were totaled for each penalty block.

#### 2.3.9. Soccer Equipment

The equipment used for the penalty kick sessions: a goal box (7.32 m × 2.44 m) on a game-regulated size field with a size 5 ball and distance to the penalty mark (11 m) were in accordance with NCAA regulations [24]. The penalty kick was taken within the goal box at the designated penalty spot on the field.

### 2.4. Procedure

Before the study began, an Institutional Review Board (IRB) approval was confirmed. Participants signed the informed consent and completed the demographic questionnaire. Next, the participants were familiarized with the DELSYS Trigno Avanti sensors, HR monitor, and the penalty kick procedure. The participants were allowed to have a warm-up period of five minutes consisting of their normal warm-up routine before games/practices. The participants then completed the three blocks of five penalty kicks against a goalkeeper. The first block of penalty kicks served for baseline data. The second block represented the high-stress situation, and the third block represented the VR intervention in the high-stress situation. One minute of rest was given between each penalty kick, and a five-minute seated rest in a shaded area between each condition was provided. Researchers were the only ones present during the study with the participant. Due to the fact that the study was completed on an intramural public field, some spectators were present as they were passing by or on adjacent fields in the area, and total seclusion could not be achieved. Each block is explained further below.

#### 2.4.1. Penalty Kick Block 1: Baseline

The participants were told that the main purpose of these penalty kicks was to ensure the HR monitor and the Trigno Avanti sensors were working accordingly, aiming to relieve any anxiety in this first session. They first completed the MRF-3 questionnaire and then proceeded to take the five penalty kicks. There was a 60 s break between each penalty shot. Once the participants completed the five penalty kicks, they completed the RSME questionnaire and then rested seated in a shaded area for five minutes.

#### 2.4.2. Penalty Kick Block 2: Stress Induced with No VR Intervention

After rest, the participants were read a script that specified that they were to successfully score as many of the five shots as they could against the goalkeeper. They were told that their scores would be analyzed and totaled after the session and there would be no way to know their score, therefore, to do the best they can. They were told that their scores would be compared to those of the other participants and ranked on a scoreboard if they made the top five and to imagine that their performance would be communicated with their coach to help with playing time decisions. This method of inducing stress has been utilized in a similar study and found effective, thus it was used for the present study [18]. Participants then completed the MRF-3, penalty kicks, and RSME form in the exact same manner as in the first penalty block, and then rested seated for another five minutes.

#### 2.4.3. Penalty Kick Block 3: Stress Induced with VR Intervention

After rest, the stress-inducing script was reiterated to the participants to maintain the high-stress situation. Before they completed the penalty shots, the participants watched the relaxation intervention seated with the VR HMD. When finished, the participants completed the MRF-3, the five penalty kicks, and the RSME like in the previous two blocks. The participants then completed the commitment check. Once all forms were completed, the participants were debriefed and told that their performance did not have any effect on playing time; they were only told this to induce a stress response and were then thanked for their participation.

### 2.5. Data Analysis

Demographic information was analyzed with descriptive statistics. Kinematic analysis of the kicking movement was examined with a movement magnitude measure (peak acceleration) and a temporal (time to peak acceleration) variable. Using a lumbar sensor, movement initiation was determined as 3 standard deviations from a baseline steady state (i.e., standing still) and visually checked by a researcher. Peak accelerations were identified for both sensors within a 1.5 s window following movement initiation. Time to peak acceleration was used as the temporal measurement and calculated from movement initiation to the timestamp of peak acceleration for each sensor.

Since the data were not normally distributed, Friedman tests were carried out to compare the MRF-3 subscales, RSME, HR, penalty kick score, and peak acceleration across the three kicking conditions. Dunn–Bonferroni post hoc tests were used, and effect size was measured by Kendall’s *W*.

## 3. Results

Three Friedman tests were carried out to compare the cognitive anxiety, somatic anxiety, and confidence for the three conditions (see Figure 1). A significant difference between the conditions on cognitive anxiety was found, *χ*^2^ (2) = 18.63, *p* < 0.001. Dunn–Bonferroni post hoc tests were carried out, and the results revealed that VR was significantly different from baseline (*p* = 0.002) and stress (*p* < 0.001). Kendall’s *W* was 0.72 which indicates a large effect size. Comparison of the mean scores for each of the significant dependent variables suggested that VR (*M* = 2.15, *SD* = 1.14) reported lower scores compared to the means of baseline (*M* = 4.77, *SD* = 1.36) and stress (*M* = 5.62, *SD* = 2.14).

On the subscale of somatic anxiety, a significant difference was reported between the conditions, *χ*^2^ (2) = 21.50, *p* < 0.001. Dunn–Bonferroni post hoc tests were carried out, and the results revealed that VR was significantly different from baseline (*p* = 0.001) and stress (*p* < 0.001). Kendall’s *W* was 0.83, which indicates a large effect size. Comparison of the mean scores suggested that VR (*M* = 2.23, *SD* = 1.17) reported lower scores compared to the means of baseline (*M* = 5.46, *SD* = 1.39) and stress (*M* = 5.08, *SD* = 1.26).

On the subscale of confidence, a significant difference was reported between the conditions, *χ*^2^ (2) = 18.43, *p* < 0.001. Dunn–Bonferroni post hoc tests were carried out, and the results revealed that VR was significantly different from baseline (*p* = 0.002) and stress (*p* = 0.001). Kendall’s *W* was 0.71, which indicates a large effect size. Comparison of the mean scores suggests that VR (*M* = 3.38, *SD* = 1.12) reported lower scores (higher self-confidence) compared to the means of baseline (*M* = 5.54, *SD* = 0.78) and stress (*M* = 5.31, *SD* = 1.18).

A Friedman test was carried out to compare RMSE for the three conditions (see Figure 2). A significant difference between the conditions was found, *χ*^2^ (2) = 10.16, *p* = 0.006. Dunn–Bonferroni post hoc tests were carried out and revealed that stress was significantly different from that in baseline (*p* = 0.007). Kendall’s *W* was 0.39, which indicates a moderate effect size. Comparison of the mean scores suggests that stress (*M* = 76.23, *SD* = 33.15) reported higher scores compared to the means of baseline (*M* = 56.68, *SD* = 26.19) and VR (*M* = 57.62, *SD* = 29.19).

A Friedman test was carried out to compare penalty kick scores for the three conditions (see Figure 3). There was a nonsignificant difference between the conditions *χ*^2^ (2) = 3.11, *p* = 0.21. Kendall’s *W* was 0.12, which indicates a small effect size. Total penalty kick scores did not differ significantly across the penalty block conditions.

A Friedman test was carried out to compare HR for the three conditions (see Figure 4). A significant difference between the conditions was found, *χ*^2^ (2) = 7.96, *p* = 0.02. Dunn–Bonferroni post hoc tests were carried out and showed that VR was significantly different from baseline (*p* = 0.02). Kendall’s *W* was 0.31, which indicates a moderate effect size. Comparison of the mean scores suggests that VR (*M* = 104.08, *SD* = 16.51) reported lower HR compared to the means of baseline (*M* = 110.23, *SD* = 16.90) and stress (*M* = 107.69, *SD* = 18.44). 

Friedman test was carried out to compare acceleration amplitudes of the lumbar and thigh sensors for the three conditions (see Figure 5). Due to a data collection error, only 11 participants were analyzed. On the lumbar sensor, there was a nonsignificant difference between the conditions, *χ*^2^ (2) = 5.64, *p* = 0.06. Kendall’s *W* was 0.26, which indicates a small effect size. On the thigh sensor, there was a nonsignificant difference between the conditions, *χ*^2^ (2) = 1.27, *p* = 0.53. Kendall’s *W* was 0.06, which indicates a small effect size.

### Commitment Check

All participants actively engaged with the VR relaxation scene. All 13 participants felt that the VR helped them relax, and 11 participants (84.62%) felt that it helped them reduce anxiety. In terms of using VR again, 11 participants (84.62%) reported that they would use VR relaxation again, and the remaining two (15.38%) participants were indifferent. Regarding how participants felt about using VR before a competition, eight participants (61.54%) felt that the VR relaxation would help them before a competition to perform better. The other five participants (38.46%) either selected “no” or “indifferent”.

## 4. Discussion

The present study examined the effects of a VR relaxation intervention on anxiety and performance of penalty kicks in female soccer players. All positions were included in the present study, providing a better-balanced examination of the effectiveness of VR relaxation in female soccer players. All players on a team have the potential to take a penalty kick, and the concentration of position in the group taking penalty kicks varies from team to team. Even goalkeepers can take a penalty kick against the opposing goalkeeper, so it was important to encompass all positions. The results indicated that the VR relaxation intervention reduced cognitive and somatic anxiety while increasing confidence compared to the stress condition. All participants felt that the VR helped relax them prior to their last five penalty kicks. The VR intervention brought participants closer to their baseline levels by lowering their perceived effort while also reducing their HR. However, despite these relaxing effects, the participants penalty kicks’ scores in the VR condition were lower, although insignificantly, compared to those of the stress condition, while movement strategies were not influenced by either the stress or VR conditions.

A key research question for this study was if a VR relaxation technique would significantly reduce the perceived anxiety levels compared to the stress and baseline conditions. After the VR relaxation intervention, perceived anxiety levels were reduced significantly, while self-confidence significantly increased, showing that the VR intervention had a positive effect. These results support research from Liu and Matsumura [16], who surveyed Division-I student athletes about the relaxing effects of a VR intervention and found that the athletes reported the VR to be relaxing and beneficial to them. This relaxation effect also significantly lowered HRs in the VR condition compared to those in baseline, therefore providing a physiological indicator of the significant relaxation effect from the VR intervention.

The other main research question examined if the VR relaxation technique would help improve penalty kick performance. Despite these improvements in the participants’ perceived levels of anxiety, their performance declined (nonsignificant) in the VR condition compared to that in the stress condition. These findings contradict other studies examining induced stress on performance in that when stress has been induced on participants, their performances suffer [9,10,11,25,26]. Wilson et al. [11] found that inducing anxiety on soccer players significantly reduced shooting accuracy. Because of the results of these previous studies, we expected to have similar findings in this study. It may be possible that the nature of the task played a role in performances. For example, Wilson et al. [11] did not use a game-regulated size goal and used a much smaller version with a goalkeeper, increasing the difficulty and reducing the chances of success. Other research has used more precise, skill-based activities (putting and table tennis) where there is less degree for error as well, while this study used an activity that has a higher rate of success regardless of anxiety. Thus, anxiety may have a larger effect on more precise sports/activities, and future research should investigate different tasks.

Additionally, movement strategies were examined through kicking kinematics to determine the effect of the VR relaxation technique. The findings showed that peak acceleration amplitudes of the thigh and lumbar were similar across all conditions. It was predicted that with the increased anxiety, participants would kick faster to attempt to ensure the shot would be made, and their kicks in the VR block would return to baseline, but results indicated no change in movement strategies. It may be the case that the stressor was not strong enough to alter the participants’ movement. In this study, the stress induced was simulated and was not the same as actual competition stress. It is also possible that the participants had practiced their kick enough under stressful conditions that their movement patterns would not be impacted. With the average of approximately 13 years of experience, the participants likely had optimized their kick to what is the most functional for them, and despite being induced with stress, this learned movement had no change. Combining this with the lower level of induced stress, the participants were able to maintain their kicking patterns in the present study.

With the reduced performance in the VR condition and improved performance in the stress condition, it is possible that the stress inducer aroused the participants to their optimum arousal level, and the VR intervention under-aroused them. According to the individual zone of optimal functioning (IZOF) theory, there is a necessary level of arousal that varies from individual to individual [27]. In respect to this theory, the stress induction in the present study may have aroused participants to the “optimal” level for performance, thus helping them execute a more precise kick. Once the participants experienced the VR intervention, the VR may have reduced their arousal levels, making their kicking technique less precise and controlled, thus making them perform worse. This possibility is apparent with HR decreasing throughout the conditions. Initially, with the first condition, there was a goal to ease any nervousness the participants had while completing the research study. With performing in front of the research team who were strangers and the novelty of being a research participant, initial anxiety may have been a factor that could not be controlled for, resulting in the elevated HR and participants being over-aroused. For the VR intervention, the intervention significantly decreased the participants’ HRs from baseline due to its relaxation effect, and the participants became under-aroused, causing HR to be at its lowest. Thus, the stress condition resulted in optimal arousal with a HR higher than that of the VR condition but lower than that of the baseline condition and coinciding with the better penalty kick scores.

Additionally, the catastrophe theory may further explain the effects of performance in regard to cognitive and somatic anxiety [28]. Accordingly, cognitive anxiety directly influences performance while somatic anxiety has a smaller effect, but if both are too high, the “catastrophe” occurs and performance plummets [29,30]. In the current study, even though cognitive anxiety was higher, participants may have risen to that peak point right before the catastrophe effect occurred, resulting in their better scores in the stress condition. Even though the stress inducer was effective in increasing anxiety in the participants, it may not have induced enough anxiety to reach the catastrophe point for them.

When measuring the participants’ perceived mental effort between conditions, it was hypothesized that the VR relaxation intervention would significantly reduce the level of perceived effort exerted in the VR condition compared to that in the stress and baseline conditions. The results indicate that the stress inducer significantly increased perceived exerted effort and, although insignificantly, the VR intervention brought the participants closer to their baseline levels. The higher level of effort in participants in the stress condition could have helped maintain their performance, as explained by the processing efficiency theory. With increased worry and anxiety, individuals can exert more effort to counterbalance the aversive effects of worry to help maintain performance [31]. Participants exerted more effort in the stress condition to counteract the reduced attentional capacity from increased anxiety and maintained their performance. In the VR condition, the participants exerted effort similarly to their baseline and thus may not have put forth the effort necessary to maintain performance and their kick, as seen with the accelerometer data, to effectively manage the stressful situation.

Based on the contradictory results between perceived anxiety and performance, it may be that VR relaxation may be inappropriate to use prior to competition due to a desired level of arousal for optimal performance [32]. VR relaxation may be more appropriate following competition or throughout the competitive season to systematically modulate psychological stress and anxiety. Injury incidence rates are higher during periods of high academic or physical stress, therefore, systematically incorporating VR relaxation can serve as a buffer during these time periods [33].

This study does not come without limitations, and caution should be used when generalizing the results of this study to other realistic soccer situations. The participants were required to wear a face covering due to COVID-19, which is not worn in actual competition. Future research should revisit this study design when participants no longer have to wear a mask to eliminate its effects. This study had a small sample size that only examined female soccer players of varying skill levels. Females tend to experience stress and anxiety differently than males, thus, the benefits of VR relaxation may be different in males, and gender differences, along with different sports, should be examined [34]. This study was conducted during COVID-19 restrictions, so our participant pool was limited to individuals within the university’s community that did not virtually attend classes, thus contributing to the small sample size. This study should be re-examined when COVID-19 restrictions and the pandemic are eased to increase the utilized sample size to better extrapolate the study’s findings. Additionally, players at different skill levels may interpret anxiety differently due to the level of competition they compete at; thus, future research should examine differences in skill level and use a trained goalkeeper for participants that match the appropriate skill level to provide consistency and proper defending. Finally, since the athletes varied in skilled level and current playing status, this could have impacted the results. The participants may have been fatigued by the third condition block, which could have resulted in reduced performance. Therefore, the current results can only be generalized to the current setting and participants. Future research should address these limitations.

## 5. Conclusions

These results from the current study are insightful for practitioners when deciding when/if to utilize VR as a relaxation intervention. Caution should still be taken when applying the results of this study to one’s practice. It is still important to consider individual differences of athletes regarding their responses to stress and anxiety. VR may be a tool for those with severe pre-competition anxiety and it may be a beneficial tool after competition when it is necessary to relax. Additionally, VR relaxation can be a viable preventative tool for upcoming periods of increased stress and anxiety.

In conclusion, the purpose of this study was to examine the effects of VR relaxation on the perceived anxiety and performance in soccer players taking a series of penalty kicks. The results indicated that the VR intervention significantly reduced the participants’ perceived cognitive and somatic anxiety levels, increased self-confidence, and reduced their HR. However, future research is still needed to understand the relationship between VR relaxation, kinematics, and performance. This study serves as an initial step in establishing a basic framework for future research to build from and evaluate VR relaxation interventions in female soccer players.

## Figures and Tables

**Figure 1 sports-09-00167-f001:**
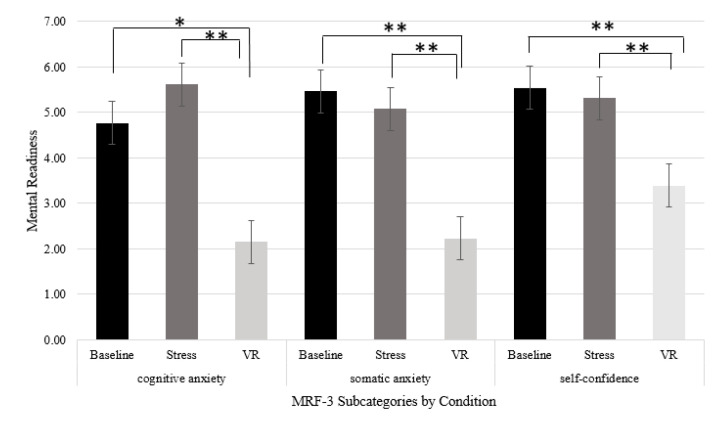
Mean MRF-3 subcategories’ ratings in penalty block conditions. Note. * = *p* < 0.05 and ** = *p* < 0.001.

**Figure 2 sports-09-00167-f002:**
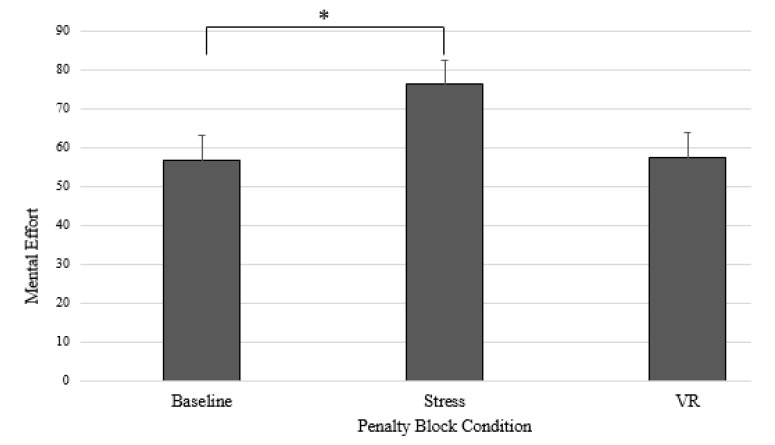
Mean RSME ratings in penalty block conditions. Note. * = *p* < 0.05.

**Figure 3 sports-09-00167-f003:**
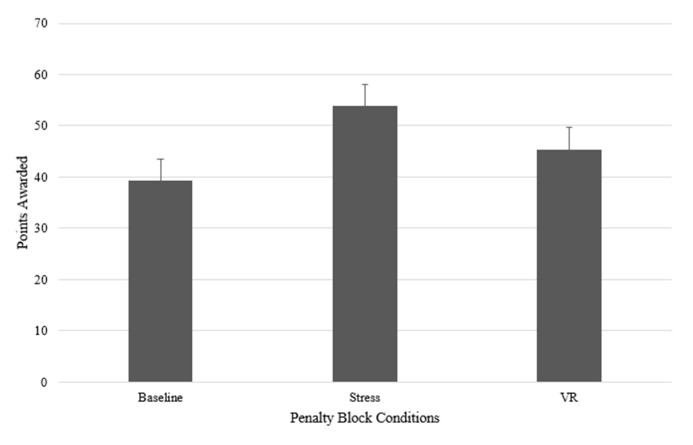
Mean penalty kick scores in the penalty block conditions.

**Figure 4 sports-09-00167-f004:**
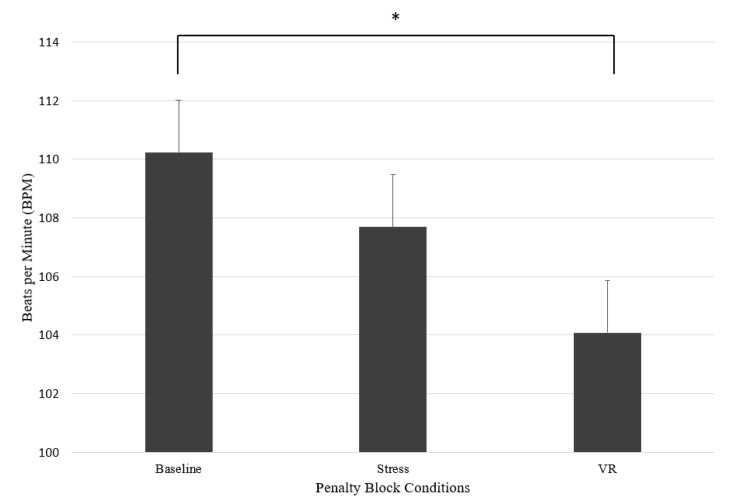
Mean heart rate (BPM) in penalty block conditions. Note. * = *p* < 0.05.

**Figure 5 sports-09-00167-f005:**
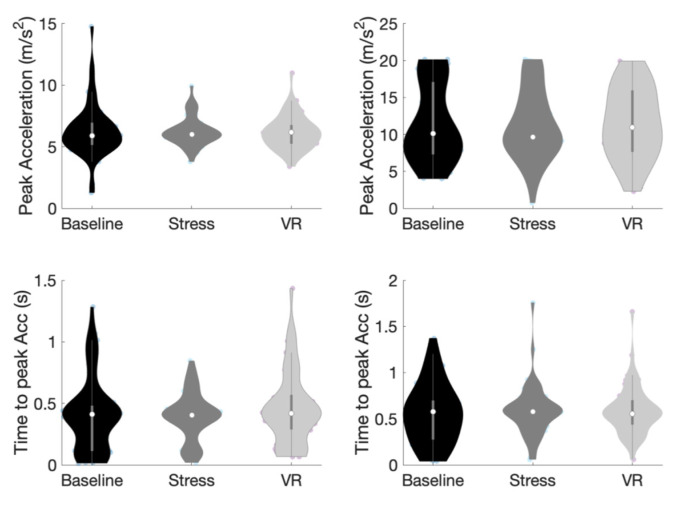
Peak acceleration (top row) and mean time to peak acceleration (bottom row) for lumbar (left column) and thigh (right column) sensors in penalty block conditions.

## Data Availability

The data presented in this study are available on request from the corresponding author.

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
