# Peer review of "The Effectiveness of Virtual Reality on Anxiety and Performance in Female Soccer Players"

_sports, 2021, doi:10.3390/sports9120167_

Round 1
Reviewer 1 Report
The authors have investigated an interesting new development within applied sport psychology / sport performance settings. The use of Virtual Reality technology is being implemented at great speed but with only limited scientific supporting evidence. As such more research is needed in this area and I commend the authors for this. With that said, the are numerous shortcomings to the current submission, which need to be addressed before publication can be considered. The two major concerns that I have are: 1) the extremely small sample size, and 2) the inability to implement a VR relaxation in the game environment. Would it be possible for the authors to extend the study and increase the participant numbers? This would allow for greater validity in the statistical analysis. Then if finding that VR can improve performance, a discussion is needed as to how to implement this approach to an applied setting.
The following specific comments are provided to assist the authors in developing the manuscript further:
Introduction:
The structure of the introduction needs to be developed. The novelty of the study is the use of VR and this needs to highlighted throughout. It may be best for the first paragraph to focus on the use of VR and how it is being used in sport / sport psychology (see (2020) The use of virtual reality head-mounted displays within applied sport psychology, Journal of Sport Psychology in Action, 11(2), 115-128 as a starting point). In subsequent paragraphs the literature on anxiety and performance could then be discussed.
L31 - a reference is required after "A contributor to psychological stress is competitive anxiety". Also expand as to why this is the case.
L41-42 - the reference to golfers could be removed.
L46 - remove 'additionally'
L58-59 - explain how VR reduces anxiety in non-athletic populations
L60 - The Liu and Matsamura (2019) study is purely related to relaxation and has no sport performance indicators. The authors may need to make note of this and the potential for differences with the current study.
L72 -why has there been no previous reference to 'accelerometer data'. Please explain why this will be helpful.
Methods:
Participants - alongside the low participant numbers, how many of these participants would regularly take penalty kicks? This may have a significant impact.
Please add full psychometric properties for the included questionnaires.
L101 - more is needed to explain why kinematics are being assessed
L114 - please explain how the physical exertion of taking each penalty is accounted for
L146 - Please include the IRB reference number
L155 - was a goalkeeper present in all scenarios? Also, please explain the performance setting (i.e., were there spectators, coaches, teammates, etc. present?)
L167-168 - is this actually stress producing? There is no relation to game setting, when penalties would be take and have more meaning. The 'treat' of reduced playing time is not convincing, especially as there are 13 participants in the study and 11 players will make up the starting line-up of the team.
Can the authors consider if stress is the only factor related to unsuccessful performance? How are skill errors, for example, accounted for?
Results:
Figure 3 demonstrates that the best performance comes from the stressful situation. So why should a VR relaxation program be used?
Figure 4 appears flawed. Can the authors explain why HR decreases at each stage
Discussion:
The discussion will need to be updated based on the above comments. Some observations on this are:
- stress activation is known to increase performance, so why would all stress be removed
- somatic anxiety can be individualised and varies between participants
- with only 13 participants can any relaxation statistical results be concluded?
Author Response
Thank you for your feedback on the manuscript. Significant revisions have been made to the manuscript based on your recommendations. These revisions are in track changes in the revised manuscript. Below are point by point descriptions of the changes in italics and red font.
The two major concerns that I have are: 1) the extremely small sample size, and 2) the inability to implement a VR relaxation in the game environment. Would it be possible for the authors to extend the study and increase the participant numbers? This would allow for greater validity in the statistical analysis. Then if finding that VR can improve performance, a discussion is needed as to how to implement this approach to an applied setting.
The authors recognize the limitation of the sample size. Given the effect size found for the main dependent variable (mental readiness) we are confident in the findings. The small effect size on kicking performance and movement kinematics is not unusual for this type of psychological study. Also, changing COVID guidelines from our institution has the potential to alter the findings if additional participants were recruited since face coverings are no longer required. Currently, there are plans in place to extend the current findings to a larger sample and under a more traditional research environment.
Yes, it is true that the VR relaxation technique used in the current study cannot be implemented within a game scenario; however, its utility still has importance for applied settings. Particularly, for players who experience high anxiety or others who have a poor ability to manage anxiety the use of VR during a practice session can provide them with necessary skills that hopefully transfer to the game setting.
Introduction:
The structure of the introduction needs to be developed. The novelty of the study is the use of VR and this needs to highlighted throughout. It may be best for the first paragraph to focus on the use of VR and how it is being used in sport / sport psychology (see Bird, J. (2020) The use of virtual reality head-mounted displays within applied sport psychology, Journal of Sport Psychology in Action, 11(2), 115-128 as a starting point). In subsequent paragraphs the literature on anxiety and performance could then be discussed.
The introduction has been modified begin with information on VR in sport and then transition into the other paragraphs. Additionally, information has been included to develop introduction on accelerometer (L96-102).
L31 - a reference is required after "A contributor to psychological stress is competitive anxiety". Also expand as to why this is the case.
A reference has been added along with an explanation as to why (L44; L47-49). When competitive anxiety becomes too high, the athlete can become over aroused, exceeding the mental capacity to process stimuli, resulting in an increase in stress and decline in performance
L41-42 - the reference to golfers could be removed.
Reference to golfers has been removed.
L46 - remove 'additionally'
‘Additionally’ has been removed
L58-59 - explain how VR reduces anxiety in non-athletic populations
It is explained that through exposure therapy to stimuli and with audio-visual experiences, anxiety in nonathletic populations has been seen to be reduced with both strategies (L79-84)
L60 - The Liu and Matsamura (2019) study is purely related to relaxation and has no sport performance indicators. The authors may need to make note of this and the potential for differences with the current study.
It was noted that the study purely focused on relaxation, but their survey results about the perception of VR improving performance from the participants warrants a more extensive look into the possibility of VR reducing stress to improve performance. Thus, the current study may have different results due to investigating the relationship of VR relaxation and performance (L87-93).
L72 -why has there been no previous reference to 'accelerometer data'. Please explain why this will be helpful.
It was explained why accelerometer data was utilized and how it is helpful (L96-103). This study utilized accelerometer data as the physical component to examine if induced stress and/or the VR intervention affected participants’ technique/motor pattern and track any changes across the duration of the study. Due to the novelty of this study and it serving as an initial step in this direction of research, accelerometer data was utilized for its ease of use and simplicity in measuring biomechanical changes out on a soccer field.
Methods:
Participants - alongside the low participant numbers, how many of these participants would regularly take penalty kicks? This may have a significant impact.
There is not a specific position that is designated to take penalty kicks. Coaches usually decide on who takes penalty kicks for tie breakers and/or penalties by observing all players taking penalty kicks in practice. Even goalkeepers can take penalty kicks against the opposing team’s goalkeeper if the coach chooses to do so. Thus, it varies from team to team on position concentration for the group that takes the penalty kicks in a penalty shootout. So, all participants would have a potential for taking penalty kicks. They are not used too often in the sport but can be the most stress-inducing part of the game, which is why they are used for this study.
Please add full psychometric properties for the included questionnaires.
Psychometric properties were added for the questionnaires.
L101 - more is needed to explain why kinematics are being assessed
More information was added on accelerometer data (L160-166)
L114 - please explain how the physical exertion of taking each penalty is accounted for
Physical exertion of taking each penalty kick was not directly measured by researchers due to the fact that a penalty kick does not require much energy compared to what the rest of the sport requires. However, we still considered fatigue as a possibility with participants, so a rest period of 1 minute between each kick, and then 5 minutes between each condition was implemented to ensure proper recovery if fatigue occurred. We also provided them a chair to sit during their rest between each condition and access to water so over exertion was avoided. This has been clarified in the manuscript (L 229-230).
L146 - Please include the IRB reference number
As per the journal’s guidelines, the IRB reference number is listed at the end of the manuscript.
L155 - was a goalkeeper present in all scenarios? Also, please explain the performance setting (i.e., were there spectators, coaches, teammates, etc. present?)
It is now specified that a goalkeeper was used in each scenario as well as the performance setting (L225; L229-232). Researchers were the only ones present during the study with the participant. The study was completed on an intramural public field.
L167-168 - is this actually stress producing? There is no relation to game setting, when penalties would be take and have more meaning. The 'treat' of reduced playing time is not convincing, especially as there are 13 participants in the study and 11 players will make up the starting line-up of the team.
This method of inducing stress through scoring and comparison has been used in a prior similar study and was found to be sufficient in inducing stress, thus why it is used for the present study. This is now stated in the manuscript (L248-250).
Can the authors consider if stress is the only factor related to unsuccessful performance? How are skill errors, for example, accounted for?
It is mentioned in the limitations that the skill level/competition level the participants play at may have a role in their performance scores and how they manage competitive anxiety, thus why future research should investigate this possibility further and recruit from a highly advanced population. To limit as much skill error as possible, we recruited from a pool that had prior soccer experience before to try to mitigate this limitation. Skill error was also attempted to be documented with the accelerometer data, but specifically with the impact of stress and relaxation and not skill error in general.
Results:
Figure 3 demonstrates that the best performance comes from the stressful situation. So why should a VR relaxation program be used?
This is described in the discussion section (see lines 489-492). VR relaxation may be inappropriate to use prior to competition due to a desired level of arousal for optimal performance [27]. VR relaxation may be more appropriate following competition or throughout the competitive season to systematically modulate psychological stress and anxiety.
Figure 4 appears flawed. Can the authors explain why HR decreases at each stage
It has been added in the discussion with the IZOF section as to why HR may have decreased with each stage (L456-465).
Discussion:
The discussion will need to be updated based on the above comments. Some observations on this are:
stress activation is known to increase performance, so why would all stress be removed
somatic anxiety can be individualised and varies between participants
with only 13 participants can any relaxation statistical results be concluded?
Future research and the continuation of this study will be able to provide a sounder conclusion on how relaxation plays a role in performance. It was not intended to remove all stress in the present study, but was something discovered that VR may do, thus why it may be more harmful, which is discussed as well. There are cases where too much stress occurs resulting in a poor performance, which is more common than a lack of stress, so the findings of the present study were not something we expected to see. It is also mentioned that with those with severe competitive anxiety, VR may be beneficial due to how powerful it may potentially be and is something that future research should consider and needs to be considered by practitioners before use.

Reviewer 2 Report
Dear Authors:
I congratulate you for the work. It is a very interesting research. Perhaps the main problem is the small sample with which they work. However, you should make a series of modifications:
1. You should change the title, "Effects" to "Effectiveness".
2. In the abstract they should adequately describe the sample, include the main numerical results and insert the size of the effect.
3. They should conduct a more thorough review of the literature. I recommend a review of these papers:
-Ortigosa-Márquez J.M., Carranque-Cháves G.A., and Hernández Mendo, A. (2015). Effects of autogenic training on lung capacity, competitive anxiety and sub-jective vitality. Biomedical Research, 26 (1), 71-76.
-Reigal, Rafael E.; Vázquez-Diz, Juan A.; Morillo-Baro, Juan P.; Hernández-Mendo, Antonio; Morales-Sánchez, Verónica. (2020). Psychological Profile, Competitive Anxiety, Moods and Self-Efficacy in Beach Handball Players. Int. J. Environ. Res. Public Health 17, (1): 241. https://doi.org/10.3390/ijerph17010241.
-Reigal, R.E.; Páez-Maldonado, J.A.; Pastrana-Brincones, J.L.; Morillo-Baro, J.P.; Hernández-Mendo, A.; Morales-Sánchez, V. (2021). Physical Activity Is Related to Mood States, Anxiety State and Self-Rated Health in COVID-19 Lockdown. Sustainability, 13, 5444. https://doi.org/10.3390/su13105444 2071-1050.
4. Before the Participants section, they should identify the type of research design and the reference that supports it.
5. They should provide the complete reference of the ethics committee. Confirm whether the study was conducted in accordance with the Declaration of Helsinki (WMA 2000, Bošnjak 2001, Tyebkhan 2003), which sets out the fundamental ethical principles for research involving human subjects.
6. They should perform a prior analysis of normality through the Shapiro-Willks test for small samples.
7. Before using the MANOVA test, they should justify the use of this test with small samples, which probably does not have a normality fit.
8. You must calculate the effect size.
9. They should perform a generalizability analysis with the objective of demonstrating that the sample allows considering that the results obtained are reliable and generalizable.
I believe that once these changes have been made, the work will be ready for publication.
Yours sincerely.
Round 2
Reviewer 1 Report
The authors have revised the manuscript to a high standard and this version is suitable for publication.
Author Response
Thank you for your feedback.
Reviewer 2 Report
I think the authors have solved almost all the issues. However there is one issue that is not solved:
"They should perform a generalizability analysis with the aim of demonstrating that the sample allows the results obtained to be considered reliable and generalizable."
The authors have replied:
"We are not clear what is meant by a generalizability analysis. The analyses reveal large effect sizes for the main dependent variable (mental preparedness). It has been clarified in the discussion that these data can only be generalized to this group and setting (L514- 515). "
I think it is important that they perform these analyses in order to ensure that the results obtained with this sample are reliable and generalizable. It would be a good closure for this work. I recommend that you review the following papers:
-Rubén Maneiro, Ángel Blanco-Villaseñor and Mario Amatria (2020). Analysis of the Variability of the Game Space in High Performance Football: Application of the Generalizability Theory. Front. Psychol. , March 25, 2020 | https://doi.org/10.3389/fpsyg.2020.00534。
-Reigal, R.E.; González-Guirval, F.; Pastrana-Brincones, J.L.; González-Ruiz, S.; Hernández-Mendo, A.; Morales-Sánchez, V. (2020). Analysis of Reliability and Generalizability of One Instrument for Assessing Visual Attention Span: MenPas Mondrian Color. Sustainability, 12, 7655.
